# PROSE: A Pronoun Omission Solution for Chinese-English Spoken Language Translation

**Ke Wang**[†]  **Xiutian Zhao**[†]  **Yanghui Li**  **Wei Peng**[*]

Huawei IT Innovation and Research Center

{wangke215, zhaoxiutian, liyanghui, peng.wei1}@huawei.com

## Abstract

Neural Machine Translation (NMT) systems encounter a significant challenge when translating a pro-drop ('pronoun-dropping') language (e.g., Chinese) to a non-pro-drop one (e.g., English), since the pro-drop phenomenon demands NMT systems to recover omitted pronouns. This unique and crucial task, however, lacks sufficient datasets for benchmarking. To bridge this gap, we introduce PROSE, a new benchmark featured in diverse pro-drop instances for document-level Chinese-English spoken language translation. Furthermore, we conduct an in-depth investigation of the pro-drop phenomenon in spoken Chinese on this dataset, reconfirming that pro-drop reduces the performance of NMT systems in Chinese-English translation. To alleviate the negative impact introduced by pro-drop, we propose Mention-Aware Semantic Augmentation, a novel approach that leverages the semantic embedding of dropped pronouns to augment training pairs. Results from the experiments on four Chinese-English translation corpora show that our proposed method outperforms existing methods regarding omitted pronoun retrieval and overall translation quality.

## 1 Introduction

In recent years, neural machine translation (NMT) technology has made significant progress in lowering communication barriers between individuals from different language backgrounds. However, NMT systems often struggle when translating sentences from a pro-drop ('pronoun-dropping') language, such as Chinese, Korean and Japanese (Shimazu et al., 2020), to a non-pro-drop language, such as English, French and German (Haspelmath, 2001)). While the pro-drop phenomenon has been widely studied in the research community (Nagard

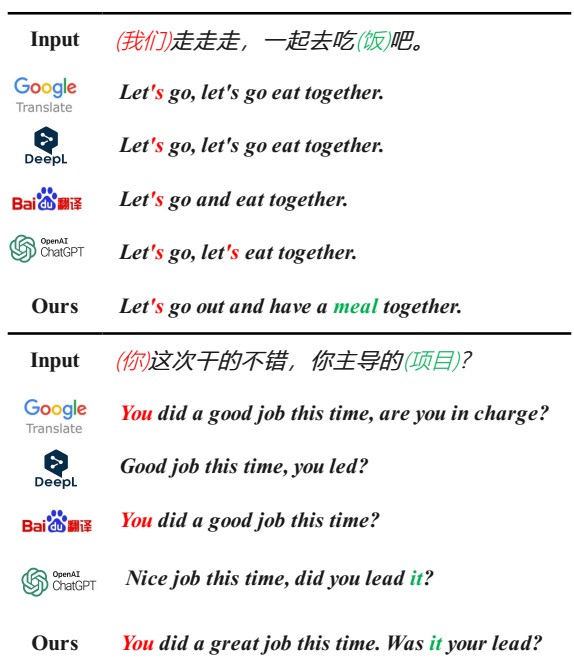

Figure 1: Examples of pro-drop in daily spoken Chinese with corresponding English translations provided by our model and several mature commercial NMT systems (Google, DeepL, Baidu and ChatGPT, respectively, data collected on January 13th, 2023). The Chinese pronouns in brackets are dropped, and the dropped subject and object pronouns are marked in red and green, respectively.

and Koehn, 2010; Taira et al., 2012; Wang et al., 2016, 2018a; Tan et al., 2019), advanced commercial NMT systems occasionally fail to faithfully recover dropped pronouns in the source language. In some cases, leaving missed pronouns unrecovered could result in severe semantic distortion and alter the intended meaning of the translated text, as demonstrated in Figure 1.

To tackle this issue, researchers have proposed two primary strategies: (1) incorporating additional pro-drop resolution systems to provide supplementary syntactic information (Nagard and Koehn, 2010; Taira et al., 2012; Wang et al., 2016). For instance, Xiang et al. (2013) modeled *Empty*

---

[†]Equal Contribution.
[*]Corresponding author.

*Categories* within the framework of government-binding theory; (2) treating pro-drop resolution as a regularization component of NMT task directly (Wang et al., 2018b; Tan et al., 2019). This approach suggests filling dropped pronouns (Wang et al., 2018a) or predicting pro-drops in the Chinese text encoder component of a seq2seq model (Wang et al., 2019). Despite the studies done on resolving Chinese pro-drop in NMT so far, relevant benchmarks evaluating the effectiveness of pro-drop mitigation are highly limited, and Chinese-English spoken translation datasets with fine-grained annotation are even fewer.

In this study, we present PROSE, a **PR**onoun **O**mission **S**olution for Chinese-English spoken language translation. To facilitate research in this area, we introduce a novel dataset for document-level Chinese-English spoken language translation that includes abundant and diverse pro-drop instances with contextual and pro-drop annotations across four spoken language genres (talk, drama, movie, and vlog). The analysis of this dataset reveals that the negative impact of pro-drop on Chinese-English spoken language translation. Furthermore, we propose the Mention-Aware Semantic Augmentation approach, which utilizes a mention encoder to capture the semantic embedding of dropped pronouns and employs a mention-side data augmentation technique to generate additional training pairs. Experiment results on four Chinese-English translation corpora demonstrate that our proposed approach significantly increase translation quality and the recover rate of missed pronouns, in comparison with baseline methods on both automatic and human evaluation metrics. Additionally, we conducted ablation studies to provide further insights on the effect of designated losses.

Our contributions are summarized as follows:

- We construct a document-level Chinese-English spoken translation dataset that covers multiple spoken genres and provides detailed contextual and pro-drop annotation information.

- Our analysis reveals that pro-drop negatively impacts the quality of Chinese-English spoken language translation.

- We propose a Mention-Aware Semantic Augmentation approach to increase the recover rate of dropped pronouns when translating and hence enhance overall translation quality.

## 2 Dataset Creation

To mitigate the scarce of benchmarks evaluating pro-drop in Chinese-English spoken language translation, we collect and construct a new benchmark, PROSE, a high-quality Chinese-English bilingual dataset of four different genres, including *Talk*, *Drama*, *Movie* and *Vlog*.

### 2.1 Data Collection and Filtering

The raw data was collected from bilingual subtitles of publicly accessible videos on the internet. We assume that these subtitles reflect authentic daily spoken expressions in Chinese and cover a diverse range of zero anaphora phenomena. Specifically, our filtering process is based on three criteria.

- The chosen domain must be spoken, rather than written, such as news articles, to preserve the colloquial features of Chinese;

- To ensure high-quality English translations, we only considered source materials in Chinese that have undergone manual translation by professionals, rather than relying on machine translations. For instance, we primarily chose movies from China that have been promoted overseas and short videos with professional fan-made translations on YouTube.;

- To enable accurate restoration of missing pronouns, the source material must contain contextual sentences that provide greater context and accuracy to the translations.

We end up collecting over 20,000 videos in Chinese and over 2 million lines of English and Chinese subtitle, which can be classified into four distinct spoken genres:

- **Talk:** Subtitles from personal presentations on websites like TED.

- **Drama:** Subtitles from Chinese TV series, such as *Nirvana in Fire* (琅琊榜).

- **Movie:** Subtitles from Chinese films, such as *The Great Cause of the Founding of the People* ( 建国大业 ).

- **Vlog:** Subtitles from short videos filmed by Chinese internet celebrities, such as *Huanong Brothers* (华农兄弟).

## 2.2 Pro-drop Annotation

We employ DDparser (Zhang et al., 2020), a Chinese dependency parsing tool, to detect the omissions of subject or object pronouns in the source language Chinese. Subject pronouns are tagged as SBV (subject-verb) or VV (verb-verb), while object pronouns are tagged as VOB (verb-object), POB (preposition-object), DOB (double object), or DBL (double). Dependencies that do not contain these marks are assumed to be missing either the subject or object pronoun. Although this method of labeling is not perfect, it warrants further study. Below is an example from the subtitles of a short video about cooking.

> *Chinese:* 四伯爷这个从哪儿下刀哦.
> *English:* Uncle, where should I start cutting? *[Subject Ellipsis]*
>
> *Chinese:* 咋个下刀. *[Subject Ellipsis] [Object Ellipsis]*
> *English:* How do I start cutting this?
>
> *Chinese:* 那个肉留不留在上面吗. *[Subject Ellipsis]*
> *English:* Will you leave the meat on the bone?

As shown, each data pair consists of Chinese text with its corresponding pronoun type missing, high-quality English translations done by human experts, and the surrounding context. We apply the DDparser tool on the training set to annotate whether there is *Subject Ellipsis* and *Object Ellipsis* in the Chinese sentences, while the English sentences require no annotation. This is due to that we only collected source materials in Chinese that have undergone manual translation by professionals. The high-quality translations have completed the subject and object in English. For the test set, in addition to calculating the BLEU score with human translation, we also use human evaluation to assess Completeness, Semantic Correctness, and Overall quality (details can be found in Appendix C).

We randomly sample 100 samples and manually check the accuracy of the *Subject Ellipsis* and *Object Ellipsis* marked by the annotation tool. The experimental results are shown in the table 1.

## 2.3 Data Statistics

The data statistics for our datasets, which include four genres of spoken Chinese, are presented

| Accuracy | Talk | Drama | Movie | Vlog |
|---|---|---|---|---|
| Subject Ellipsis | 93.4% | 89.4% | 90.4% | 85.6% |
| Object Ellipsis | 95.3% | 90.3% | 91.4% | 87.3% |

Table 1: The accuracies of the *Subject Ellipsis* and *Object Ellipsis* marking by the annotation tool.

in Table 2. CWMT2018[1] is the most popular Chinese-English machine translation corpus, containing written language such as news articles, while AIChallenger[2] is the largest spoken Chinese-English machine translation dataset to the best of our knowledge.

In comparison with those two widely used bilingual datasets, our dataset is 1) more representative with a higher pro-drop ratio, 2) more diverse, containing four genres of spoken language, and 3) more informative, with contextual and pro-drop annotation information.

## 3 Pronoun-Dropping Analysis

To gain more insights into the phenomenon of pro-drop in the translation of spoken Chinese into English, we examine the prevalence of pro-drop in spoken Chinese and its impact on the quality of Chinese-English spoken language translation.

**Spoken Chinese Contains More Pro-drop Than Literary Language** Formally, pro-drop refers to a reference position that is filled with amorphologically unrealized form, and is one of the most common referential options in many languages such as Chinese (Wang et al., 2018a), Japanese (Taira et al., 2012), Korean (Park et al., 2015), and Thai (Kawtrakul et al., 2002). Previous studies have revealed that spoken Chinese language tends to contain more pro-drops than literary language (Wang et al., 2016, 2017; Xu et al., 2021). However, quantitative studies on pro-drops in different genres of spoken Chinese, remain scarce.

As demonstrated in Table 2, both written and spoken languages contain a certain proportion of pro-drops, which is consistent with the unique grammatical phenomenon of Chinese. However, written language contains fewer *Object Ellipsis* than spoken language. For example, in the CWMT2018 dataset, the proportion of *Object Ellipsis* (i.e., 2.80%) is significantly smaller than that of *Subject Ellipsis* (i.e., 9.00%). Our four bilingual spoken language corpora are varied, displaying differences in the

---

[1] http://nlp.nju.edu.cn/cwmt-wmt/
[2] https://github.com/AIChallenger/AI_Challenger_2018

| Datasets | Spoken | Type | #Doc. | #Sen. | #Sent./#Doc. | English #Word / #Sent. | Chinese #Word / #Sent. | Subject Ellipsis | Object Ellipsis |
|---|---|---|---|---|---|---|---|---|---|
| **CWMT2018** | ✗ | - | ✗ | 9,023,454 | - | 11.54 | 19.65 | 9.00% | 2.80% |
| **AIChallenger** | ✓ | - | ✗ | 8,426,940 | - | 11.54 | 17.99 | 8.10% | 8.90% |
| **Talk** | ✓ | Train | 1,613 | 193,965 | 120.25 | 17.38 | 29.35 | 9.11% | 8.01% |
| | | Test | 179 | 21,325 | 119.13 | 17.29 | 29.79 | 9.16% | 7.92% |
| **Drama** | ✓ | Train | 22,499 | 2,150,956 | 95.60 | 5.90 | 8.86 | **32.77%** | **33.03%** |
| | | Test | 25 | 2,240 | 89.60 | 8.31 | 9.35 | **40.40%** | **36.52%** |
| **Movie** | ✓ | Train | 134 | 108,162 | 807.18 | 5.66 | 9.53 | **37.38%** | **35.84%** |
| | | Test | 14 | 13,137 | 938.36 | 5.50 | 10.88 | **35.78%** | **35.21%** |
| **Vlog** | ✓ | Train | 667 | 75,051 | 112.52 | 6.67 | 8.95 | **45.91%** | **37.45%** |
| | | Test | 74 | 7,536 | 101.84 | 6.85 | 9.21 | **45.38%** | **33.80%** |

Table 2: The data distribution of our Chinese-English pro-drop machine translation datasets. *Doc.* and *Sen.* indicate *Document* and *Sentence* respectively. # stands for the quantity, and / denotes the ratio.

rates of subject and object pronoun drop, average sentence length, average document length, and so on. For example, the average length of sentences in the three genres of spoken corpora, namely Drama, Movie and Vlog, is much shorter than that of Talk (i.e., individual talks) and AIChallenger. In particular, the Drama, Movie and Vlog corpora in our data set contain a surprising proportion of pro-drops (about 33% to 46%), which is more extensive than the current largest Chinese-English spoken translation corpus AIChallenger.

**Pro-drop Harms the Quality of Chinese-English Spoken Language Translation**  Subjective and objective pronouns are frequently omitted in spoken Chinese, but should be recovered in non-pro-drop languages like English. The question arises whether the current NMT system is able to accurately translate spoken Chinese sentences with dropped pronouns into English, a non-pro-drop language, as illustrated in Figure 1.

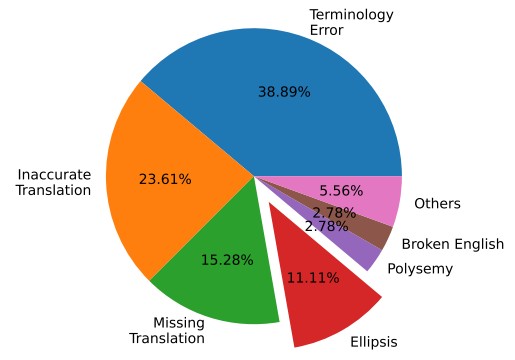

Figure 2: Error distribution of Chinese-English Spoken translation in our online simultaneous translation system. Errors caused by pro-drop (i.e, Ellipsis) account for about 11% of errors.

Figure 2 shows the distribution of Chinese-to-

English translation errors in our online simultaneous machine translation system. The primary use case of our translation system is Chinese-to-English translation (primarily spoken Chinese) in meetings and conferences. Moreover, we have experienced labeling experts to categorize the bad cases generated by the online system. It can be seen from Figure 2 that the proportion of errors caused by pro-drop is relatively high, constituting more than 11% of all errors. This is one of the major factors contributing to the degradation of the translation quality of our system.

| Dataset | Zero Pronoun | Trans-former | +Human Completion | Δ |
|---|---|---|---|---|
| **CWMT** | 9.0% | 27.40 | 27.91 | +0.51 |
| **Talk** | 9.2% | 15.56 | 16.31 | +0.75 |
| **Drama** | 40.4% | 13.38 | 15.68 | +2.30 |
| **Movie** | 35.8% | 14.92 | 16.62 | +1.70 |
| **Vlog** | 45.3% | 9.38 | 12.25 | +2.87 |

Table 3: Results of Chinese-English spoken language translation with the omitted pronouns complemented by humans. Although the model achieved a high BLEU score of 27.40 on the CWMT dataset, its performance on their dataset showed a significant decline, with a BLEU score dropping from 9.38 to 15.56.

To investigate the potential of reinstated pronouns in Chinese spoken sentences to improve the quality of Chinese-English spoken language translation, we conduct experiments using spoken Chinese sentences with omitted pronouns complemented by humans. We first train a Transformer-base (Vaswani et al., 2017; Hassan et al., 2018) model on the CWMT dataset, and then report the BLEU (Papineni et al., 2002) scores with Sacre-BLEU[3] (Post, 2018) on test sets of our four cor-

---

[3] 4BLEU + case.mixed + lang.LANGUAGE PAIR + num-

pora (i.e., Talk, Drama, Movie and Vlog). Next, the spoken Chinese in test sets that is detected as pro-drop are completed manually, as shown in the content in brackets in Figure 1.

The experimental results before and after human completion are shown in Table 3. Although the model achieves a 27.40 BLEU score on the CWMT dataset, its performance on our dataset shows a significant BLEU score decline (from 9.38 to 15.56 across four genres). This indicates a large discrepancy between spoken and written Chinese for neural machine translation systems that rely on data-driven approaches. For convenience, the second column in Table 3 displays the proportion of different datasets with pro-drop. Human completion of dropped pronouns leads to varying performance improvement levels, with the improvement being roughly proportional to the ratio of pro-drops. Interestingly, even on the CWMT dataset, human completion has improved translation quality (i.e., +0.51 BLEU score ), suggesting that pro-drop may also degrade the quality of the Chinese-English translation of that dataset.

## 4 Methodology

### 4.1 Problem Definition

Given two data spaces, $\mathcal{X}$ and $\mathcal{Y}$, encompassing all possible sentences in source (Chinese) and target (English) languages, each sample is a pair of sentences belonging to different languages, i.e., $(\boldsymbol{x}, \boldsymbol{y}) \in (\mathcal{X}, \mathcal{Y})$. Here, $\boldsymbol{x} = \{x_1, x_2, \cdots, x_{|\boldsymbol{x}|}\}$ is the Chinese sentence containing $|\boldsymbol{x}|$ tokens, and $\boldsymbol{y} = \{y_1, y_2, \cdots, y_{|\boldsymbol{y}|}\}$ is the English sentence with $|\boldsymbol{y}|$ tokens. To identify the mentions (co-references) of entities (i.e., pronouns) in $\boldsymbol{x}$, its surrounding context is expressed as $\boldsymbol{c}$. For example, in the context of $\boldsymbol{c} =$ "饭应该做好了 (The meal should be ready)", the missing object pronoun of "吃 (eat)" in the sentence $\boldsymbol{x} =$ "走走走，一起去吃吧" can be inferred to be "饭 (meal)", thus the translation of the non-pro-drop sentence would be *"Let's go out and have a **meal** together"*.

The neural machine translation task (Bahdanau et al., 2015; Gehring et al., 2017; Vaswani et al., 2017) seeks to model the translation probability $P(\boldsymbol{y}|\boldsymbol{x}, \boldsymbol{c}; \Theta)$ using a conditional language model based on Transformer architecture (Vaswani et al., 2017), where $\Theta$ represents the parameters of the model to be optimized. Formally, the training objective of a given set of observed sentence pairs is

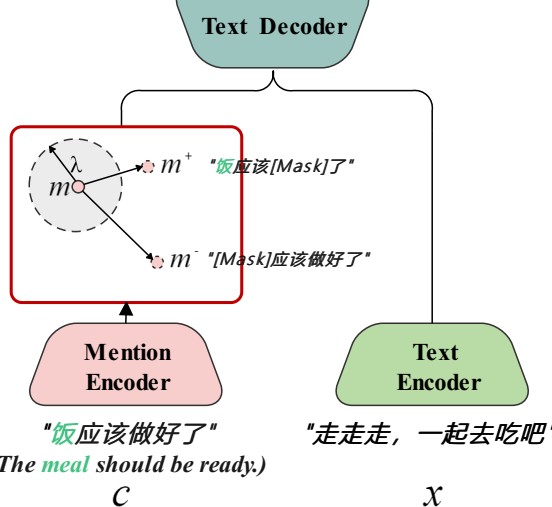

Figure 3: The framework of mention-aware semantic augmentation. $\boldsymbol{x}$ and $\boldsymbol{y}$ represent sentences in the source and target languages, respectively. The contextual text is denoted by $\boldsymbol{c}$.

to maximize the log-likelihood:

$$\mathcal{L}_{nmt}(\Theta) = \mathbb{E}_{(\boldsymbol{x}, \boldsymbol{y}) \sim (\mathcal{X}, \mathcal{Y})}(\log P(\boldsymbol{y}|\boldsymbol{x}, \boldsymbol{c}; \Theta)). \quad (1)$$

### 4.2 Mention-Aware Semantic Augmentation

Motivated by the high prevalence of pro-drop in spoken Chinese and the consequent difficulty in automatically understanding pro-drop source sentences when translated into non-pro-drop English, we present Mention-Aware Semantic Augmentation (illustrated in Figure 3) as a potential solution.

**Architecture** This approach is built on top of Transformer (Vaswani et al., 2017) and consists of three modules: a text encoder $E_t$, a text decoder $D_t$, and an additional mention encoder $E_m$. The mention encoder $E_m$ is a 6-layer transformer encoder which translates the context $\boldsymbol{c}$ to representations $E_m(\boldsymbol{c}) \in \mathbb{R}^k$, where $k$ is the embedding dimension. To obtain a real-valued output, a projection matrix $\boldsymbol{A} \in \mathbb{R}^{k \times k}$ is applied to $E_m(\boldsymbol{c})$, resulting in $\boldsymbol{m} = E_m(\boldsymbol{c})\boldsymbol{A}$. The mention representation $\boldsymbol{m}$ and the text representation $\boldsymbol{r}$ are concatenated together at each time-step and sent to the decoder to calculate the cross attention. It is worth noting that our mention encoder module shares parameters with the text encoder $E_t$. Moreover, it is agnostic to the model architecture and can easily be adapted to other text generation frameworks.

Overall, our approach leverages 1) the mention encoder to focus on completing the dropped pronouns in the input $x$ from the context $c$ in the case of limited parallel corpus, and 2) representation interpolation in the semantic space of observed samples to expand the training data pairs, thus compensating for the lack of large-scale Chinese-English spoken language translation corpora.

**Mention-Aware Contrastive Learning**   We propose a contrastive objective to learn the semantic embeddings $m$ of mentions in the source sentence $x$. Specifically, the representations of sentences containing mentions of entities should be "closer" to $m$ than those without mentions.

To this end, we expect the similarity between $m$ and a "similar" sample $m^+$ to be far greater than that between $m$ and a negative sample $m^-$, i.e., $Sim(m, m^+) \gg Sim(m, m^-)$. To obtain $m^-$, we use DDparser (Zhang et al., 2020) to detect all mentioned entities in the context, and then randomly replace them with a special token [MASK]. $m^+$ is sampled by randomly replacing non-entity words. The measure of similarity between two embeddings, denoted as $Sim$, is calculated using the dot product. This can be interpreted as the angle between the two embeddings in the vector space. Consequently, the mention-aware contrastive objective is formulated as follows:

$$\mathcal{L}_{mcl}(\Theta) = -\mathbb{E}_{(\boldsymbol{x},\boldsymbol{y})\sim(\mathcal{X},\mathcal{Y})}[ \qquad (2)$$
$$\log \frac{\exp(Sim(\boldsymbol{m}, \boldsymbol{m}^+))}{\exp(Sim(\boldsymbol{m}, \boldsymbol{m}^+)) + \exp(Sim(\boldsymbol{m}, \boldsymbol{m}^-))}].$$

We introduce a regularization loss to further reduce the disagreements among the mention projection matrix and reduce the redundancy of parameters: $\mathcal{L}_{reg}(\Theta) = ||\boldsymbol{A}^T\boldsymbol{A} - \boldsymbol{I}||^2$, where $\boldsymbol{I}$ is the identity matrix.

**Mention-Side Mixup Interpolation**   Drawing inspiration from Mixup approaches (Zhang et al., 2018; Wang et al., 2021; Wei et al., 2022), we propose to sample data points from the adjacency mention semantic region to augment the current training instance. Given pairs of samples $(x_1, y_1)$ and $(x_2, y_2)$, Mixup chooses a random mixing proportion $\lambda$ from a Beta distribution $\beta(\alpha, \alpha)$ controlled by the hyper-parameter $\alpha$, and creates an artificial training example $(\lambda x_1 + (1-\lambda)x_2, \lambda y_1 + (1-\lambda)y_2)$ to train the network by minimizing the loss on mixed-up data points:

$$\mathcal{L}_{mix}(\Theta) = \mathbb{E}_{x_1,y_1 \sim p_\mathcal{D}} \mathbb{E}_{x_2,y_2 \sim p_\mathcal{D}} \mathbb{E}_{\lambda \sim \beta(\alpha,\alpha)}[ \quad (3)$$
$$\ell(\lambda x_1 + (1-\lambda)x_2, \lambda y_1 + (1-\lambda)y_2)],$$

where $\ell$ is the cross entropy loss (de Boer et al., 2005). According to **Appendix A**, we can simplify Equation 3 as follows:

$$\mathcal{L}_{mix}(\Theta) \Rightarrow \mathbb{E}_{x_1,y_2 \sim p_\mathcal{D}} \mathbb{E}_{x_2 \sim p_\mathcal{D}} \mathbb{E}_{\lambda \sim \beta(\alpha+1,\alpha)} \quad (4)$$
$$\ell(\lambda x_1 + (1-\lambda)x_2, y_1),$$

which enables us to avoid the requirement for label blending when combining labels $y_1$ and $y_2$, with $\lambda$ drawn from $\beta(\alpha + 1, \alpha)$. This is beneficial in scenarios where $y_2$ is a discrete sequence. Accordingly, our mention-side mixup loss minimizes the interpolations loss from a vicinity distribution (Chapelle et al., 2000) defined in the representation space:

$$\mathcal{L}_{mmi}(\Theta) = \mathbb{E}_{(\boldsymbol{x}_i,\boldsymbol{y}_i)\sim(\mathcal{X},\mathcal{Y})} \mathbb{E}_{\lambda \sim \beta(\alpha+1,\alpha)} \quad (5)$$
$$(\log P(\boldsymbol{y}_i | \boldsymbol{x}_i, \lambda\boldsymbol{m}_i + (1-\lambda)\boldsymbol{m}_i^+); \Theta)).$$

In other words, we can utilize the presence or absence of pronoun context (i.e, $m$ and $m^+$) to argument the training samples for enhancing the robustness towards pronouns.

### 4.3   Training and Inference

Finally, we optimize the sum of the above losses:

$$\mathcal{L}_{final}(\Theta) = \mathcal{L}_{nmt}(\Theta) + \mathcal{L}_{mcl}(\Theta) \qquad (6)$$
$$+ \mathcal{L}_{reg}(\Theta) + \mathcal{L}_{mmi}(\Theta).$$

During inference, beam search decoding is performed.

## 5   Experiments

### 5.1   Baselines Comparisons

We compare our method with several state-of-the-art machine translation methods, including pro-drop machine translation methods (**RecNMT** (Wang et al., 2018a) and **pro-dropP&T** (Wang et al., 2019)), document-level machine translation methods (**HanNMT** (Miculicich et al., 2018)), and data-augmentation machine translation methods (**AdvAug** (Cheng et al., 2020) and **CsaNMT** (Wei et al., 2022)). We pre-train the NMT model using the AIChallenger dataset, achieving 27.97 BLEU

| Method | Type | | | Dataset | | | | Average | Δ |
|---|---|---|---|---|---|---|---|---|---|
| | pro-drop? | DL? | DA? | Talk | Drama | Movie | Vlog | | |
| **Base** | ✗ | ✗ | ✗ | 13.10 | 11.99 | 16.09 | 5.09 | 11.57 | - |
| **Fine-tuning** | ✗ | ✗ | ✗ | 16.41 | 17.29 | 17.73 | 13.89 | 16.33 | +4.76 |
| **RecNMT** (Wang et al., 2018a) | ✓ | ✗ | ✗ | 17.46 | 17.97 | 18.06 | 13.98 | 16.89 | +0.53 |
| **pro-dropP&T** (Wang et al., 2019) | ✓ | ✗ | ✗ | 17.97 | 17.98 | 18.12 | 14.27 | 17.09 | +0.76 |
| **HanNMT** (Miculicich et al., 2018) | ✗ | ✓ | ✗ | 18.92 | 18.92 | 19.05 | 17.62 | 18.63 | +2.30 |
| **AdvAug** (Cheng et al., 2020) | ✗ | ✗ | ✓ | 18.82 | 18.12 | 18.89 | 14.86 | 17.67 | +1.34 |
| **CsaNMT** (Wei et al., 2022) | ✗ | ✗ | ✓ | 18.58 | 18.32 | 19.32 | 17.08 | 18.33 | +2.00 |
| **Ours** | ✓ | ✓ | ✓ | **19.46** | **19.87** | **20.34** | **18.47** | **19.54** | **+3.21** |

Table 4: Automatic evaluation results on our Chinese-English spoken language translation dataset. The acronym *"pro-drop?"* stands for *"pro-drop?"*, *"DL?"* for *"Document-Level?"*, and *"DA?"* for *"Data-Augmentation?"*.

points on the test set. Afterward, we optimize the parameters on our specific spoken Chinese corpus, which is relatively small in size. The implementation details are shown in **Appendix B**.

For analysis, we also show the performance of NMT models trained on different corpora, including: (1) **Base**: Training the NMT model solely on a small Chinese-English spoken language translation corpus. (2) **Fine-tuning**: Training the NMT model on the AIChallenger dataset and then fine-tuning the model on Chinese-English spoken corpora.

## 5.2 Automatic Evaluation

For automatic translation evaluation, we report the classical BLEU (Papineni et al., 2002) scores with SacreBLEU. The automatic evaluation results on our four-genre Chinese-English spoken translation dataset are presented in Table 4.

Our experiment results show that Fine-tuning method outperforms the Base method by 4.76 BLEU points, indicating that the amount of data remains the bottleneck of translation performance on the task of spoken language translation with a limited corpus. Furthermore, the document-level machine translation method (HanNMT) is significantly better than single-text-input-based methods (RecNMT and pro-dropP&T) and data-augmentation-based methods (AdvAug and CsaNMT), indicating that context information is useful for pro-drop translation. Interestingly, the data-augmentation-based NMT methods (AdvAug and CsaNMT) also have an approximate BLEU gain of 1.34 to 2.00, demonstrating that the sampling method in the semantic space to expand the training dataset can well enhance the generalization of pro-drop spoken language translation. In any case, our method greatly outperforms these baseline methods, demonstrating the effectiveness of our proposed approach for pro-drop translation.

| Genre | Method | Pron. | Seman. | Overall |
|---|---|---|---|---|
| **Talk** | RecNMT | -0.32 | -0.08 | -0.45 |
| | HanNMT | -0.12 | -0.06 | -0.23 |
| | CsaNMT | 0.19 | -0.18 | 0.25 |
| | Ours | **0.25** | **0.32** | **0.43** |
| **Drama** | RecNMT | -0.79 | -0.42 | -0.11 |
| | HanNMT | **0.68** | -0.30 | -0.64 |
| | CsaNMT | -0.47 | 0.28 | 0.31 |
| | Ours | 0.58 | **0.44** | **0.44** |
| **Movie** | RecNMT | -0.23 | -0.12 | -0.01 |
| | HanNMT | 0.04 | -0.72 | -0.58 |
| | CsaNMT | -0.66 | **0.52** | 0.24 |
| | Ours | **0.85** | 0.32 | **0.35** |
| **Vlog** | RecNMT | -0.55 | -0.33 | -0.28 |
| | HanNMT | 0.53 | -0.22 | -0.94 |
| | CsaNMT | -0.53 | **0.30** | 0.44 |
| | Ours | **0.55** | 0.25 | **0.78** |

Table 5: Human evaluation results in terms of the Best-Worst scaling. The kappa coefficient of judges is 0.52.

## 5.3 Human Evaluation

We also conduct a human evaluation focusing on three metrics: **pronoun recovery** (determining whether the translated sentence is complete or contains missing mentions), **semantic correctness** (determining whether the translated sentence is semantically consistent with the source text sentence) and **overall quality**.

We sample 200 instances from four corpora, and hired two workers to rate the translation results of pro-dropP&T, HanNMT, CsaNMT and our model based on the above three aspects. We used Best-Worst Scaling, which has been shown to produce more reliable results than ranking scales (Kiritchenko and Mohammad, 2017). Specifically, each score is computed as the percentage of times it was selected as best minus the percentage of times it was selected as worst, and ranges from -1 (unanimously worst) to +1 (unanimously best). The order in which the translated texts were presented to the

| # | $\mathcal{L}_{nmt}$ | $\mathcal{L}_{mcl}$ | $\mathcal{L}_{reg}$ | $\mathcal{L}_{mmi}$ | Talk | Drama | Movie | Vlog | Average | $\Delta$ |
|---|---|---|---|---|---|---|---|---|---|---|
| 1 | ✓ | ✓ | ✓ | ✓ | 19.46 | 19.87 | 20.34 | 18.47 | 19.54 | - |
| 2 | ✓ | ✓ | ✓ |   | 18.97 | 19.28 | 19.12 | 17.92 | 18.82 | -0.72 |
| 3 | ✓ |   |   | ✓ | 18.47 | 18.57 | 18.35 | 17.52 | 18.23 | -1.31 |
| 4 | ✓ | ✓ |   | ✓ | 19.37 | 19.43 | 19.97 | 18.06 | 19.21 | -0.33 |

Table 6: Ablation study of different losses.

judges was random. The details of the questions can be found in **Appendix C**.

Table 5 indicates that HanNMT, a strong document-level machine translation method, performs better than CsaNMT and RecNMT in recovering missing pronouns, possibly due to its use of rich source-side context. Interestingly, CsaNMT, which utilizes data augmentation, exhibits superior semantic correctness and overall quality. Nonetheless, our method outperforms all baselines in terms of pronoun recovery and overall quality, indicating that the performance improvement is attributed to pro-drop resolution. More examples of generated translations of our model against comparison systems are presented in **Appendix D**.

### 5.4 Ablation Study

We conduct various ablation studies on our dataset as shown in Table 6, which assess the contribution of different losses. The SacreBLEU scores are reported on test sets.

The experiment results show that the removal of Mention-Side Mixup Interpolation results in a 0.72 BLEU point drop, indicating that the data augmentation method based on mentions can increase the generalization of pro-drop translation. Moreover, we find that all our losses, especially $\mathcal{L}_{mcl}$, are beneficial for improving the translation quality. This implies that our mention-aware contrastive learning is capable of capturing the lost pronoun information and thus improving overall performance of NMT.

It is worth noting that the third row in Table 6 is a strong document-level machine translation baseline, indicating that the improvement of our model mainly comes from the mention-aware loss rather than the wide contexts in the source side.

## 6 Related Work

**Pro-drop in Machine Translation** Research on pro-drop machine translation mainly falls into two categories: (1) methods using extra pro-drop resolution systems and (2) joint pro-drop resolution and translation training methods. The former relies on some syntax tools to provide extra information for the MT system (Nagard and Koehn, 2010; Taira et al., 2012; Wang et al., 2016), such as modeling empty categories (Xiang et al., 2013). However, directly using the results of external pro-drop resolution systems for the translation task shows limited improvements (Taira et al., 2012), since such external systems are trained on small-scale data that is non-homologous to MT. To bridge the gap between the two tasks, some later studies (Wang et al., 2018b; Tan et al., 2019; Xu et al., 2022) directly integrated the pro-drop resolution task into the machine translation task, such as reconstructing the missing pronouns (Wang et al., 2018a) in the encoder or predicting the pro-drop (Wang et al., 2019).

Unlike previous methods, our method recovers the missing pro-drop information from the context and uses data augmentation in the semantic space to increase the training data. To the best of our knowledge, we are the first to construct a document-level Chinese-English spoken translation dataset covering multiple spoken genres.

**Document-Level Machine Translation** Recent works on customized model architectures have focused on improving context representations for document-level translation models, such as context-aware encoders (Voita et al., 2019a), context-aware decoders (Voita et al., 2019b), hierarchical history representations (Miculicich et al., 2018), and memory networks (Maruf and Haffari, 2018). However, Yin et al. 2021 points out that simply feeding contextual text may not be able to accurately disambiguate pronouns and polysemous words that require contexts for resolution. Thus, we employ contrastive learning to enforce the model to incorporate the mention-information about the dropped pronouns.

**Data Augmentation in Machine Translation** Our approach is also related to Vicinal Risk Minimization (Chapelle et al., 2000), which formalizes data augmentation as extracting additional pseudo samples from the vicinal distribution of observed

instances (Krizhevsky et al., 2012; Zhang et al., 2018; Wang et al., 2021). In machine translation, this vicinity is often defined through adversarial augmentation with manifold neighborhoods (Cheng et al., 2020; Wei et al., 2022). Our approach is similar in that it involves an adjacency mention semantic region as the vicinity manifold for each training instance.

## 7 Conclusion

This study provides valuable insights into the phenomenon of pro-drop in Chinese and its impact on Chinese-English spoken language translation. Furthermore, we introduced a new dataset that improves upon existing corpora in terms of representativeness, diversity, and informational value. Lastly, our proposed approach, Mention-Aware Semantic Augmentation, demonstrates superior performance over existing methods in addressing the challenges posed by pro-drops.

Our study underscores the critical importance of taking into account pro-drops in NMT systems, and offers valuable benchmarks and insights to guide future advancements in this field.

## Limitations

Our method has shown effectiveness in improving translation quality in pro-drop machine translation task from pro-drop languages such as Chinese to a non-pro-drop target language, English in this case. However, due to the limited availability of data resources, the translation performance from other pro-drop languages such as Japanese (Sasano and Kurohashi, 2011), Thai (Kawtrakul et al., 2002), Korean (Park et al., 2015), Italian (Iida and Poesio, 2011), Spanish (Palomar et al., 2001), etc. to non-pro-drop languages remains to be evaluated. Furthermore, our method may not be able to match the performance of large language models such as PaLM (Chowdhery et al., 2022), ChatGPT[4] and GPT4[5] , which are trained with massive machine translation corpora and other language resources.

## Acknowledgement

We would like to express our deepest appreciation to Suqing Yan, Weixuan Wang, and Xupeng Meng from Huawei for their invaluable assistance and discussions during the implementation process. Their

insights and expertise have been instrumental in the realization of our ideas. We are also immensely grateful to the anonymous reviewers for their constructive feedback and comments. Their perspectives have greatly enhanced the quality of our work. Lastly, we extend our gratitude to all those who have indirectly contributed to this project. Your support has not gone unnoticed and is much appreciated.

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

# Appendix

## A  Proof of Equation 4

$$
\begin{aligned}
\mathcal{L}(\Theta) &= \mathbb{E}_{x_1,y_1\sim p_\mathcal{D}}\mathbb{E}_{x_2,y_2\sim p_\mathcal{D}}\mathbb{E}_{\lambda\sim\beta(\alpha,\alpha)}[\ell(\lambda x_1 \\
&\quad + (1-\lambda)x_2, \lambda y_1 + (1-\lambda)y_2)] \quad\quad (7)\\
&= \mathbb{E}_{x_1,y_1\sim p_\mathcal{D}}\mathbb{E}_{x_2,y_2\sim p_\mathcal{D}}\mathbb{E}_{\lambda\sim\beta(\alpha,\alpha)}[ \\
&\quad \lambda\ell(\lambda x_1 + (1-\lambda)x_2, y_1) \\
&\quad + (1-\lambda)\ell(\lambda x_1 + (1-\lambda)x_2, y_2)] \quad (8)\\
&= \mathbb{E}_{x_1,y_1\sim p_\mathcal{D}}\mathbb{E}_{x_2,y_2\sim p_\mathcal{D}}\mathbb{E}_{\lambda\sim\beta(\alpha,\alpha)} \\
&\quad \mathbb{E}_{z\sim Ber(\lambda)}[z\ell(\lambda x_1 + (1-\lambda)x_2, y_1) \\
&\quad + (1-z)\ell(\lambda x_1 + (1-\lambda)x_2, y_2)] \quad (9)\\
&= \mathbb{E}_{x_1,y_1\sim p_\mathcal{D}}\mathbb{E}_{x_2,y_2\sim p_\mathcal{D}}\mathbb{E}_{z\sim Ber(0.5)} \\
&\quad \mathbb{E}_{\lambda\sim\beta(\alpha+z,\alpha+1-z)}[z\ell(\lambda x_1+ \\
&\quad (1-\lambda)x_2, y_1) + (1-z)\ell(\lambda x_1+ \\
&\quad (1-\lambda)x_2, y_2)] \quad\quad (10)\\
&= 0.5*\mathbb{E}_{x_1,y_1\sim p_\mathcal{D}}\mathbb{E}_{x_2,y_2\sim p_\mathcal{D}}\mathbb{E}_{\lambda\sim\beta(\alpha+1,\alpha)} \\
&\quad [\ell(\lambda x_1 + (1-\lambda)x_2, y_1)] + 0.5*\mathbb{E}_{x_1,y_1\sim p_\mathcal{D}} \\
&\quad \mathbb{E}_{x_2,y_2\sim p_\mathcal{D}}\mathbb{E}_{\lambda\sim\beta(\alpha,\alpha+1)}[\ell(\lambda x_1+ \\
&\quad (1-\lambda)x_2, y_2)] \quad\quad (11)\\
&= 0.5*\mathbb{E}_{x_1,y_1\sim p_\mathcal{D}}\mathbb{E}_{x_2,y_2\sim p_\mathcal{D}}\mathbb{E}_{\lambda\sim\beta(\alpha+1,\alpha)} \\
&\quad [\ell(\lambda x_1 + (1-\lambda)x_2, y_1)] + 0.5*\mathbb{E}_{x_2,y_2\sim p_\mathcal{D}} \\
&\quad \mathbb{E}_{x_1,y_1\sim p_\mathcal{D}}\mathbb{E}_{(1-\lambda)\sim\beta(\alpha,\alpha+1)}[\ell((1-\lambda)x_2 \\
&\quad + \lambda x_1, y_1)] \quad\quad (12)\\
&\Rightarrow \mathbb{E}_{x_1,y_2\sim p_\mathcal{D}}\mathbb{E}_{x_2\sim p_\mathcal{D}}\mathbb{E}_{\lambda\sim\beta(\alpha+1,\alpha)}\ell(\lambda x_1 \\
&\quad + (1-\lambda)x_2, y_1) \quad\quad (13)
\end{aligned}
$$

- Eq (8): Linearity of the loss: $\ell(x, py_1 + (1-p)y_2) = p\ell(x, y_1) + (1-p)\ell(x, y_2)$, where the loss is the cross entropy loss.

- Eq (9): Expectation of a Bernoulli($\lambda$).

- Eq (10): The Beta distribution is conjugate prior for the Bernoulli.

- Eq (11): Expectation of a Bernoulli(0.5).

- Eq (12): Symmetry of the Beta distribution in the sense that $\lambda \sim (a, b)$ implies $1 - \lambda \sim (b, a)$.

- Eq (13): Changing variable names in the expectation.

## B  Implementation Details

We implement our method on top of the Transformer-base (Vaswani et al., 2017) implemented in Fairseq (Ott et al., 2019). For this, the

dimension $k$ was set to 512, the number of attention heads to 8, the mention encoder $E_m$, and the text encoder $E_t$ and text decoder $D_t$ to 6 layers, and the maximum sequence length to 200. The beam size of the beam search was 5. Other hyper-parameters included a dropout rate of 0.1, Adam with a learning rate of 1e-5, $\beta_1 = 0.9$, and $\beta_2 = 0.999$. To address the out-of-vocabulary problem, we apply byte-pair-encoding (BPE) vocabulary (Sennrich et al., 2016) with 40k merge operations and set $\alpha$ in $\beta(\alpha + 1, \alpha)$ to 0.1. We implemented our model using PyTorch and used 8 Tesla V100 graphic cards for training.

## C    Human Evaluation Questions

- **Completeness**: Does the translated sentence demonstrate syntactic completeness?

- **Semantic Correctness**: Is the translated sentence semantically correct?

- **Overall**: What is the overall quality of the translation?

## D    Examples of Generated Translations

Examples of generated translations of our model and comparison systems are show in Table 7, Table 8, Table 9, and Table 10.

| | |
|---|---|
| **Input** | 而另一个比较安全的选择：让你再拿一笔确切的金额。
(Gold: Your other option is a bit safe. Your just going to get some money for sure. )
不过只能拿500美金。
(Gold: I'm just going to give you 500 bucks.)
你可以把这笔钱放进皮夹或者马上花掉。
(Gold: You can stick it in your wallet and use it immediately.)
你的直觉决定好了吗？
(Gold: So see what your intuition is here.) |
| **HanNMT** | Another safer option: let you take another exact amount.
But only $500.
You can put the money in your wallet or spend it right away.
Have your instincts decided? |
| **CsaNMT** | Another safer option: let you take the exact amount again.
But only take $500.
You can put the money in your wallet or spend it right away.
Intuition decided? |
| **Ours** | Another more secure option would be to get an exact amount of money.
But it can only be up to $500.
You can either put it in a wallet or spend it right away.
Have you made your decision based on your intuition? |

Table 7: Examples of generated translations on **Talk**.

| | |
|---|---|
| **Input** | 很抱歉 要不要吃我的早餐?
(Gold: I'm so sorry. Do you want some of my breakfast? )
不了 沮丧得吃不下
(Gold: No, I'm too depressed to eat.)
过五分钟再说吧
(Gold: I'll probably eat in, like, five minutes.) |
| **HanNMT** | I'm sorry. Do you want my breakfast?
No, I'm too depressed to eat.
In five minutes. |
| **CsaNMT** | I'm sorry. Would you like some breakfast?
No, I'm too depressed to eat.
Talk about it in five minutes. |
| **Ours** | Sorry, do you want to have my breakfast?
No, I'm too upset to eat.
Let's talk about it in five minutes. |

Table 8: Examples of generated translations on **Drama**.

| | |
|---|---|
| **Input** | 是个女孩
(Gold: It is a girl)
四肢健全
(Gold: She has all the fingers and toes.)
大夫，她怎么不哭啊
(Gold: Doctor, how come she doesn't cry?) |
| **HanNMT** | It is a girl
Sound limbs
Doctor, why isn't she crying? |
| **CsaNMT** | Girl.
Sound limbs
Doctor, why isn't she crying? |
| **Ours** | She is a girl.
She has all her limbs.
Doctor, why isn't she crying? |

Table 9: Examples of generated translations on **Movie**.

| | |
|---|---|
| **Input** | 刚好我看一下蛇头鱼 好大的
(Gold: I just saw a snakehead fish, a very big one.)
跑哪里去了
(Gold: Where is it now?)
我原以为它们跑了 原来没跑 你看
(Gold: I thought they had run away, but they haven't, look!)
现在吃的好肥呀 它们
(Gold: They're really fat now.) |
| **HanNMT** | I'm just looking at the snakehead. It's big.
Where'd you go?
I thought they got away. They didn't. Look.
They're so fat now. |
| **CsaNMT** | Just so I can see the snakehead. It's big.
Run where?
I thought they were gone, but they weren't.
I'm so fat now. |
| **Ours** | I just took a look at the snakehead fish, it's so big.
Where did it go?
I thought it had run away, but it didn't, look.
They are eating well now, aren't they? |

Table 10: Examples of generated translations on **Vlog**.