# OpenReview forum: "PROSE: A Pronoun Omission Solution for Chinese-English Spoken Language Translation"
_EMNLP/2023/Conference — EMNLP 2023 Main_

### Official Review · Reviewer_NDb2 · 2023-07-31

**Soundness:** 4

**Excitement:**

3: Ambivalent: It has merits (e.g., it reports state-of-the-art results, the idea is nice), but there are key weaknesses (e.g., it describes incremental work), and it can significantly benefit from another round of revision. However, I won't object to accepting it if my co-reviewers champion it.

**Paper Topic And Main Contributions:**

In some languages such as Chinese and Japanese, we can remove pronoun from an utterance. Translation from a sentence with the ellipsis on pronouns to a language without such ellipsis is a challenging issue in the machine translation field. To address this issue, this paper provides a new dataset for document-level Chinese-English spoken language translation because such ellipsis frequently occurs during speaking. In addition, the authors applied contrastive learning (and mixup) to a mention encoder which encodes a context sentence. Experimental results show that the proposed method achieved better performance than several previous methods.

**Reasons To Accept:**

This paper provides a document-level Chinese-English translation dataset to address the ellipsis in translation. The dataset is useful for the machine translation research community.

**Reasons To Reject:**

I cannot agree that the proposed method is effective yet. In my understanding, several methods such as Base can use only single sentence as an input but the authors should use concatenated two sentences as an input like the baseline of [Voita et al., 18] for a fair comparison.

Voita et al., 18: Context-Aware Neural Machine Translation Learns Anaphora Resolution.

**Reproducibility:**

4: Could mostly reproduce the results, but there may be some variation because of sample variance or minor variations in their interpretation of the protocol or method.

**Reviewer Confidence:**

4: Quite sure. I tried to check the important points carefully. It's unlikely, though conceivable, that I missed something that should affect my ratings.

---

> ### Author Rebuttal · Authors · 2023-08-29
>
> Thank you for your constructive feedback! We are glad that you found our dataset useful for the machine learning research community. Below are our responses to specific comments.
>
> ***
> > I cannot agree that the proposed method is effective yet. In my understanding, several methods such as Base can use only single sentence as an input but the authors should use concatenated two sentences as an input like the baseline of [1] for a fair comparison.
>
> We thank the reviewer for pointing out this concern. In fact, we have compared our method with state-of-the-art **document-level machine translation models**, such as the HANNMT method as shown in Table 3, which includes context sentences as input. Additionally, we have demonstrated the effectiveness of our method by including a variant that only uses Mention-Side Mixup Interpolation to concatenate context without Mention-Aware Contrastive Learning in our ablation study results (Table 5, row 3).
>
> We also argue that the **effect of context is inconclusive**, as introducing context may positively or negatively impact the accuracy of the results. To further investigate this effect, we have supplemented our experiment with a simple baseline method, Fine-tuning (w/ context), where we use concatenated two sentences as input. To be more specific, the method takes the last two sentences as additional context, but the output is still only the translation of the current source language input into the target language.
>
> |           Method          |  Talk | Drama | Movie |  Vlog | Average |
> |:-----------------------:|:-----:|:-----:|:-----:|:-----:|:-------:|
> |       Fine-tuning       | 16.41 | 17.29 | 17.73 | 13.89 |  16.33  |
> | Fine-tuning(w/ context) | 16.75 | 17.08 | 17.97 | 13.93 |  16.43  |
> |           Ours          | 19.46 | 19.87 | 20.34 | 18.47 |  19.54  |
>
> The experimental results in above table suggest that simply concatenating context sentences does not provide significant benefits (an average BLEU score increase of merely 0.10) on average. Specifically, the with-context fine-tuning yielded a worsen performance (16.75) in Drama domain compared to the no-context fine-tuning (17.29).
>
> We hope this addresses your concern and provides a fair comparison of our proposed method with existing methods. Thank you for your valuable feedback.
>
> [1] Voita et al. 2018. Context-Aware Neural Machine Translation Learns Anaphora Resolution.
>
> ***
>
> Thank you for all the great suggestion! Please let us know if you have any remaining concerns, or if you would consider updating your evaluation based on our response.

---

### Official Review · Reviewer_x45u · 2023-08-01

**Soundness:** 4

**Excitement:**

4: Strong: This paper deepens the understanding of some phenomenon or lowers the barriers to an existing research direction.

**Missing References:**

none

**Paper Topic And Main Contributions:**

This paper introduces PROSE, a new dataset for document-level Chinese-English spoken language translation, focusing on pro-drop instances. To alleviate the negative impact of pro-drop, this paper also proposes Mention-aware semantic augmentation method to deal with the dropped pronouns. Experimental results confirm the effectiveness of the proposed method.

* a new dataset for document-level Chinese-English spoken language translation, focusing on pro-drop instances
* the Mention-aware semantic augmentation method to deal with pro-drop phenomena


**Questions For The Authors:**

Could you also provide the performance of LLMs, such as GPT-3.5/4?

**Reasons To Accept:**

* a new dataset for document-level Chinese-English spoken language translation, focusing on pro-drop instances, which benefits the future research on this area.
* the Mention-aware semantic augmentation method to deal with pro-drop phenomena is trained jointly with NMT framework, the contrastive learning objective and mention-side mix-up interpolation objective can inspire other research areas as well.
* have done comprehensive experiments
* well written and organized

**Reasons To Reject:**

* as the authors pointed out in the limitation section, the proposed method maybe is still inferior to LLM models, which is a key challenge in MT.
* the dataset is limited to Chinese-English language pair, and there are many other language pairs with pro-drop languageds.
* the kappa value of human evaluation in Table 4 is only 0.52 which is not that convincing.

**Reproducibility:**

4: Could mostly reproduce the results, but there may be some variation because of sample variance or minor variations in their interpretation of the protocol or method.

**Reviewer Confidence:**

4: Quite sure. I tried to check the important points carefully. It's unlikely, though conceivable, that I missed something that should affect my ratings.

**Typos Grammar Style And Presentation Improvements:**

* Table 1: use commas to seperate figures larger than 1,000. (#doc and #sen. columns)
* caption of Table 2 can be further improved: point out the ratio and the metric in the caption.
* since-wide --> source-side

---

> ### Author Rebuttal · Authors · 2023-08-29
>
> Thank you for your positive and constructive comments! We are glad that you found our dataset benefiting future research and our proposed method inspiring. Below are our responses to specific comments.
>
> ***
> > As the authors pointed out in the limitation section, the proposed method maybe is still inferior to LLM models, which is a key challenge in MT.
>
> > Could you also provide the performance of LLMs, such as GPT-3.5/4?
>
> Yes, we have supplemented our experiment by adding the performances of LLMs on our dataset. Below is a table of the BLEU score results (cf. Table 4).
>
> | Method |  Talk | Drama | Movie |  Vlog | Average |
> |:------------------:|:-----:|:-----:|:-----:|:-----:|:-------:|
> | gpt-3.5-turbo-0301 | 23.12 | 21.23 | 22.44 | 20.82 |  21.90  |
> |     gpt-4-0314     | 23.72 | 22.27 | 22.27 | 20.82 |  22.27  |
> |        Ours        | 19.46 | 19.87 | 20.34 | 18.47 |  19.54  |
>
> The results show that although the BLEU scores of GPT-3.5/4 are better than our method, the gap is not so significant. There is still a lot of room for further improvement and we will explore incorporating our method with LLM in future work. We have added the LLMs results to Table 4 in our revised version.
>
> ***
>
> > The dataset is limited to Chinese-English language pair, and there are many other language pairs with pro-drop languages.
>
> We agree with you that there are more language pairs with pro-drop languages. Besides Chinese, typical pro-drop languages include Spanish, Italian and Japanese, while English, German and French are non-pro-drop languages [1, 2]. Although a dependency parser is a prerequisite of our method, we believe that the basic idea of the method is transferable and can be generalized on other language pairs.
>
> [1] Mihoko Zushi. 2003. Null arguments: the case of Japanese and Romance.
> [2] Gereon Müller. 2005. Pro-Drop and Impoverishment.
>
> ***
>
> > The kappa value of human evaluation in Table 4 is only 0.52 which is not that convincing.
>
> We agree with you that the kappa value of 0.52 is less than good. While we are short in time conducting a new human evaluation during the rebuttal period, we will re-evaluate the results with a larger evaluator size and report additional inter-rater agreement metrics in our revised version.
>
> ***
>
> > Typos and Writing suggestions:
>
> Thank you for catching those typos and unclear expressions! We have carefully corrected them.
>
> ***
> Thank you for all the great insight and suggestions! Please let us know if you have any remaining concerns.

---

### Official Review · Reviewer_DfAP · 2023-08-05

**Soundness:** 3

**Excitement:**

3: Ambivalent: It has merits (e.g., it reports state-of-the-art results, the idea is nice), but there are key weaknesses (e.g., it describes incremental work), and it can significantly benefit from another round of revision. However, I won't object to accepting it if my co-reviewers champion it.

**Missing References:**

1. GuoFeng: A Benchmark for Zero Pronoun Recovery and Translation, EMNLP 2022

**Paper Topic And Main Contributions:**

The present paper focuses on the issue of "pronoun-dropping" in Chinese-English translation. It introduces a dataset with careful pro-drop annotation. This dataset comprises four domains: talk, drama, movie, and vlog. Additionally, the paper proposed a novel method to address the pro-drop problem, which is effectively demonstrated using the aforementioned dataset. The results of experiments conducted on four Chinese-English translation corpora reveal that the proposed method outperforms existing approaches regarding translation quality and the appropriate reinsertion of omitted pronouns.

**Reasons To Accept:**

1. The paper is well-written and easy to follow.

2. The paper released a dataset for Chinese-English translation to address the problem of pronoun-dropping.

3. This released dataset contains 4 topics: talk, drama, movie, and vlog with detailed pro-drop annotation and analysis.

**Reasons To Reject:**

1. The difference between the proposed dataset and the related work is poorly discussed. For example, [1] is also proposed for the zero pronoun translation in Chinese-English translation.

2. The annotation process (Section 2.2 Pro-drop Annotation) is not written in detail.  1) How to label the English sentence, word alignment information or any else?  2) What's the accuracy of the Chinese label process?  3) How many types of labels are used? e.g.,  [Subject Ellipsis].  I think the authors should make it clear in the following version.


[1] GuoFeng: A Benchmark for Zero Pronoun Recovery and Translation, EMNLP 2022

**Reproducibility:**

3: Could reproduce the results with some difficulty. The settings of parameters are underspecified or subjectively determined; the training/evaluation data are not widely available.

**Reviewer Confidence:**

3: Pretty sure, but there's a chance I missed something. Although I have a good feel for this area in general, I did not carefully check the paper's details, e.g., the math, experimental design, or novelty.

---

> ### Author Rebuttal · Authors · 2023-08-29
>
> Thank you for your review! We are glad that you found our dataset carefully annotated and our method novel. Below are our responses to specific comments.
>
> ***
>
> >The difference between the proposed dataset and the related work is poorly discussed. For example, [1] is also proposed for the zero pronoun translation in Chinese-English translation.
>
> We thank the reviewer for pointing out the related paper [1]. We have carefully examined the paper and decided not to include it in comprison simply because the **data and code [1] claims to release are not available**. The stated GitHub open-source address in the paper is invalid, and we are unable to obtain the dataset publicly to the best of our effort.
>
> Supposing the data of [1] is available to public, our dataset differs from the one claimed in [1] by both **size and covered domains**. The size of our dataset (25,205 documents with 2,572,372 sentences) is significantly larger than the one proposed in [1] (457 documents with 8093 sentences). [1] covers *Movie Subtitles*, *Q&A Forum*, *Government News*, *Web Fiction*, and *Personal Profile*, while our dataset covers *Talk and Interview*, *TV Drama*, *Movie*, and *Vlog*. The scope of our dataset is pro-drop in **spoken language** and all of our sources are spoken ones, while 72.8% of sentences in [1] are from **written text** sources. In addition, this study also proposes a new method, Mention-Aware Semantic Augmentation, to mitigate pro-drop errors in MT.
>
> As demonstrrated in Table 1 and discussed at Lines 189-202, we have compared 2 widely used biliginual datasets, CWMT2018 and AIChallenger. We have further augmented our analysis and comparison in line with your comment.
>
> [1] Wang et al. 2022. GuoFeng: A Benchmark for Zero Pronoun Recovery and Translation.
>
> ***
>
>
> >The annotation process (Section 2.2 Pro-drop Annotation) is not written in detail. 1) How to label the English sentence, word alignment information or any else? 2) What's the accuracy of the Chinese label process? 3) How many types of labels are used? e.g., [Subject Ellipsis]. I think the authors should make it clear in the following version.
>
> 1) As we stated in Section 2.2, we applied the DDparser tool on the training set to annotate whether there is *Subject Ellipsis* and *Object Ellipsis* in the Chinese sentences, while the English sentences require no annotation. This is due to that we only collected source materials in Chinese that have undergone manual translation by professionals. The high-quality translations have completed the subject and object in English, as described at [Lines 131-138]. For the test set, in addition to calculating the BLEU score with human translation, we also use manual evaluation to assess Completeness, Semantic Correctness, and Overall quality (details can be found in Appendix C).
>
> 2) We have randomly sampled 100 samples and manually checked the accuracy of the Subject Ellipsis and Object Ellipsis marked by the annotation tool. The experimental results are shown in the table below:
>
> |      Accuracy     |  Talk | Drama | Movie |  Vlog | Average |
> |:----------------:|:-----:|:-----:|:-----:|:-----:|:-------:|
> | Subject Ellipsis | 93.4% | 89.4% | 90.4% | 85.6% |  89.7%  |
> |  Object Ellipsis | 95.3% | 90.3% | 91.4% | 87.3% |  91.1%  |
>
> The accuracy of the annotation tool is strong. We have added this statistics in our revised version.
>
> 3) Two types of lables (*Subject Ellipsis* and *Object Ellipsis*) were used.
>
> ***
>
> Thank you for all the great questions and suggestions! Please let us know if you have any remaining concerns, or if you would consider updating your evaluation based on our response.

---

### Meta-Review · Area_Chair_8NWc · 2023-09-15

**Recommendation:** 4

**Metareview:**

The paper addresses the issue of "pronoun-dropping" in Chinese-English translation.
A dataset with careful pro-drop annotation is created, consisting of four domains: talk, drama, movie, and vlog.
The paper also proposes a novel method to address the pro-drop problem, and the results show that it outperforms existing approaches.

A few revisions are necessary, which are already provided in authors' response.

---

### Decision · Program_Chairs · 2023-10-07

**Decision:**

Accept-Main

**Comment:**

The paper addresses the issue of "pronoun-dropping" in Chinese-English translation.
A dataset with careful pro-drop annotation is created, consisting of four domains: talk, drama, movie, and vlog.
The paper also proposes a novel method to address the pro-drop problem, and the results show that it outperforms existing approaches.

A few revisions are necessary, which are already provided in authors' response.